

# Fixation patterns in pairs of facial expressions—preferences of self-critical individuals

Bronislava Šoková, Martina Baránková and Júlia Halamová

Institute of Applied Psychology, Faculty of Social and Economic Sciences, Comenius University, Bratislava, Slovakia

## ABSTRACT

So far, studies have revealed some differences in how long self-critical individuals fixate on specific facial expressions and difficulties in recognising these expressions. However, the research has also indicated a need to distinguish between the different forms of self-criticism (inadequate or hated self), the key underlying factor in psychopathology. Therefore, the aim of the current research was to explore fixation patterns for all seven primary emotions (happiness, sadness, fear, disgust, contempt, anger, and surprise) and the neutral face expression in relation to level of self-criticism by presenting random facial stimuli in the right or left visual field. Based on the previous studies, two groups were defined, and the pattern of fixations and eye movements were compared (high and low inadequate and hated self). The research sample consisted of 120 adult participants, 60 women and 60 men. We used the Forms of Self-Criticizing and Self-Reassuring Scale to measure self-criticism. As stimuli for the eye-tracking task, we used facial expressions from the Umeå University Database of Facial Expressions database. Eye movements were recorded using the Tobii X2 eye tracker. Results showed that in highly self-critical participants with inadequate self, time to first fixation and duration of first fixation was shorter. Respondents with higher inadequate self also exhibited a sustained pattern in fixations (total fixation duration; total fixation duration ratio and average fixation duration)—fixation time increased as self-criticism increased, indicating heightened attention to facial expressions. On the other hand, individuals with high hated self showed increased total fixation duration and fixation count for emotions presented in the right visual field but did not differ in initial fixation metrics in comparison with high inadequate self group. These results suggest that the two forms of self-criticism - inadequate self and hated self, may function as distinct mechanisms in relation to emotional processing, with implications for their role as potential transdiagnostic markers of psychopathology based on the fixation eye-tracking metrics.

# INTRODUCTION

## Biases in emotional face processing

Individual characteristics play an important role in eye movement and gaze fixations on human emotions. Individuals exhibit different subjective preferences regarding fixation position when viewing human faces and these individual preferences seem to persevere over

Corresponding author
Bronislava Šoková,
strnadelova1@uniba.sk

time (*Peterson & Eckstein, 2013*). *McEwan et al. (2014)* noted that self-critical individuals perceive even altruistic expressions such as a happy smile or compassion as unpleasant, rather than as supportive, warm feelings. In line with the transdiagnostic construct of self-criticism, the following studies confirmed the assumption of specific biases in the target (clinical or psychopathological) sample of respondents when viewing facial emotions.

A study of *Sears et al. (2011)* investigated attention differences in individuals with a self-reported history of depression/dysphoria/or no history of depression. Participants' eye fixations were tracked as they looked at sets of depression-related, anxiety-related, positive, and neutral images. Dysphoric individuals spent significantly less time attending to positive images than individuals with no history of depression. In addition, previously depressed individuals exhibited the same attentional bias and attended to anxiety-related images more than individuals with no history of depression, which may indicate bias in some specific clinical samples. *Allard & Yaroslavsky (2019)* showed that slow disengagement from sad faces, and rapid disengagement from happy faces was associated with brooding tendencies, a maladaptive form of rumination that is a marker of depression. In line with these findings, eyetracking revealed a double attention bias for positive and negative emotional faces in individuals with clinical depression in a study by *Duque & Vázquez (2015)*. Attentional biases in clinical, unmedicated depressed and in never-depressed participants were compared as they viewed emotional faces (happy, sad, angry) and neutral faces. The results suggest a double bias in the depressed sample (greater attention to sad faces and less attention to happy faces). Compared to never-depressed participants, participants with Major Depressive Disorder showed a negative attentional bias in first fixation duration and total fixation time on sad faces. Furthermore, the Major Depressive Disorder group spent marginally less time looking at happy faces compared with the never-depressed participant group. The use of this type of emotional expression as the stimuli appears to be important as well, because no differences were found between the groups regarding angry faces and orientation of participant attention.

Investigating clinical psychological variables, *Shechner et al. (2012)* used eye-tracking methodology to measure attention biases in anxious and non-anxious youths. The participants were shown angry, happy and neutral faces. The anxious youths displayed a greater attention bias towards angry faces compared to the non-anxious youths. Specifically, in anxious youths first fixation was more likely to occur on angry faces and fixation on angry faces was faster than on neutral faces, which could indicate some kind of preparation for a potentially threating stimulus (angry face). *Wang & Xie (2011)* studied participants' eye movements in high *versus* low trait anxiety groups using eyetracking. The results showed a similar pattern among anxious individuals—the high trait anxiety group were more sensitive to negative facial expressions than positive ones. A meta-analysis (*Armstrong & Olatunji, 2012*) of the eye-tracking research on anxiety and depression highlights subtle differences in attentional biases for both constructs—anxiety and depression. Results showed that anxiety, but not depression, is characterised by attention focusing on the threat. On the other hand, depression is characterised by greater attention to sad stimuli, and depressed individuals look less at pleasant stimuli (*Armstrong & Olatunji, 2012*). It has been demonstrated that, compared to healthy individuals, clinically depressed participants

show an attentional bias towards sad faces (*e.g.*, *Duque & Vázquez, 2015*). In a depressed sample of respondents, hypervigilance regarding negative images/faces has been interpreted mainly as evidence of insensitivity to reward, *e.g.*, other people exhibiting pleasant, kind, happy facial expressions (*Armstrong & Olatunji, 2012*).

To sum up, the literature review suggests that participants with depressive/anxious, inadequate feelings towards themselves tend to prefer stimuli or emotional expressions that reflect their attentional biases (*e.g.*, *Armstrong & Olatunji, 2012*; *Duque & Vázquez, 2015*; *Shechner et al., 2012*). It is possible that self-critical individuals may exhibit a similar pattern.

## Self-criticism and eye-tracking

*Blatt & Homann (1992)* underline the consistency of self-criticism as a form of self-concept that involves constant and powerful self-judgment and a chronic fear of others' criticism, disagreement, and rejection. Similarly, *Straub (1990)* describes self-criticism as the feeling of being criticised, imprisoned in the feeling of not being good enough and not having the energy for new things and challenges in life. People tend to engage in self-criticism for various reasons. For example, *Gilbert et al. (2004)* report that self-critical individuals either have the need to constantly improve their behaviour, or to punish themselves, to wound themselves for errors and shortcomings, and this correlates negatively with the ability to self-soothe, reassure, and focus on the positive aspects of the situation. *Falconer, King & Brewin (2015)* sees self-criticism as a stable tendency to engage in negative self-evaluation accompanied by feelings of shame. They add that it is a complex set of cognitive, emotional, motivational, and behavioural responses to the self with specific patterns.

According to *Gilbert et al. (2004)*, self-criticism manifests in two distinct forms. The first form, termed "inadequate self", involves an overestimation of errors coupled with a compulsion for correction. The second form, known as "hated self", encompasses a need for self-punishment and is characterized by feelings of contempt and hatred towards oneself. Inadequate feelings are characterised by the individual remembering and comparing their own setbacks and this is accompanied by disappointment. It is based on the belief that other people are better and results from the person comparing themselves, using their own high standards, with others (*Gilbert et al., 2004*). Hated self is focused on destructive self-critical emotions such as anger or disgust with oneself (*Gilbert et al., 2004*). This more destructive kind of self-criticism captures a crueller, disgust-based response to setbacks characterised by strong self-dislike. Beyond the two established forms of self-criticism (*Gilbert et al., 2004*), self-criticism can be operationalised in various ways. For normative populations, the recommendation is to use the combined score of self-criticism, which includes both the "hated self" and "inadequate self" forms (*Halamová et al., 2018*). Accordingly, some studies have utilised the summed raw score of the inadequate and hated self subscales from the Forms of Self-Criticizing/Attacking & Self-Reassuring Scale (FSCRS) to measure self-criticism (*Halamová et al., 2023*).

In a recent study, *Halamová et al. (2023)* investigated the relationship between self-criticism (measured by the summed score) and fixation durations on nine facial emotions using a $3 \times 3$ face-in-the-crowd paradigm. The findings revealed that highly self-critical

participants fixated on angry faces for a significantly shorter time than other participants. This result contrasts with prior research indicating an attentional bias towards negative stimuli in individuals with disorders related to self-criticism, such as depression and anxiety (*Gilbert et al., 2012*).

Given the established strong connection between self-criticism, depression, and anxiety (*Gilbert et al., 2008*), incorporating eye-tracking methodologies in research focusing on depression and anxiety could elucidate the eye-tracking patterns associated with self-criticism. Attention biases have predominantly been found towards negative stimuli (*e.g., Duque, Sanchez & Vazquez, 2014*; *Shechner et al., 2012*; *Suslow, Junghanns & Arolt, 2001*), suggesting a potential area for further exploration.

Additionally, highly self-critical people exhibited an attention avoidance tendency in relation to most emotions (*Halamová et al., 2023*). Another finding regarding recognition of facial expressions showed that a sample with a higher hated self score (more pathological than inadequate self) had difficulty identifying emotion expressions (*Strnádelová, Halamová & Kanovský, 2019*). Similarly, in self-criticism research, *Strnádelová, Halamová & Mentel (2019)* found that higher pathological self-criticism—hated self—was associated with greater avoidance of the eye region when looking at happy facial expressions, to which self-critical people are very sensitive (*McEwan et al., 2014*). Increased hated self also indicated a greater tendency to look beyond the face in photographs of multiple emotional faces (*Strnádelová, Halamová & Kanovský, 2019*). However, for inadequate self, the results did not confirm an avoidance tendency when scanning or recognising emotional faces. The findings for inadequate self (*Strnádelová, Halamová & Mentel, 2019*) indicated that this self-critical form tended to correlate positively with total fixation duration time on all the facial areas examined, although none of these correlations were statistically significant. Therefore, the tendency among self-critical people to avoid particular areas of facial expressions applies mainly in relation to hated self, although the overall fixation metrics on the expressions have not yet been sufficiently investigated. So far, we have reported the eye-tracking patterns of self-critical individuals. The research showed the need to distinguish between the different forms of self-criticism (inadequate and hated self). In addition, the studies focused on the overall metric of fixation duration (total fixation duration) when looking/avoiding looking at eye area stimuli (*e.g., Halamová et al., 2023*; *Strnádelová, Halamová & Mentel, 2019*; *Strnádelová, Halamová & Kanovský, 2019*).

To sum up, previous studies have shown that individuals with high levels of self-criticism exhibit distinct patterns of attention and interpretation when processing social and emotional stimuli. For instance, *Whelton & Greenberg (2005)* found that self-critical individuals are more likely to focus on negative aspects of facial expressions, which could be differentially influenced by whether the self-criticism stems from feelings of inadequacy or self-hatred.

Eye-tracking research (*Halamová et al., 2023*) has indicated that highly self-critical individuals exhibit different eye-tracking patterns when viewing facial expressions. Specifically, participants with high self-criticism scores, combining both "inadequate self" and "hated self", showed shorter fixation durations on angry faces, contrary to the attention bias for negative stimuli commonly found in constructs associated with

self-criticism, such as depression and anxiety (*Gilbert et al., 2012*). This avoidance of negative stimuli, particularly in hated self, suggests a need for further investigation into the distinct eye-tracking behaviors associated with each form of self-criticism.

Highly self-critical individuals also tended to avoid most emotions (*Halamová et al., 2023*), had difficulty in identifying emotional expressions (*Strnádelová, Halamová & Kanovský, 2019*), or tended to avoid a particular area of an emotional face (*Strnádelová, Halamová & Mentel, 2019*). In contrast, those with higher inadequate self scores did not show the same avoidance tendencies, but rather a general increase in total fixation duration on all facial areas, although these correlations were not statistically significant (*Strnádelová, Halamová & Mentel, 2019*).

This differentiation between inadequate self and hated self in terms of eye-tracking patterns highlights the importance of identifying the unique influences of each form of self-criticism on the perception and processing of facial expressions. Hated self appears to drive avoidance of specific emotional stimuli, particularly negative and socially evaluative expressions, reflecting the intense self-directed hostility and discomfort with social evaluation. In contrast, inadequate self may involve more general vigilance and concern about personal shortcomings, leading to prolonged scrutiny without specific avoidance.

## Aim of the study

A negative relationship with the self, characterised by high levels of self-criticism, is one of the most important psychological processes that influences the predisposition to psychopathology and its persistence (*Falconer, King & Brewin, 2015*), especially as it is related to depression and anxiety (*Gilbert et al., 2008*). In this study, we explored both fixation patterns in self-critical individuals as well as the differences between the two subscales of self-criticism—inadequate and hated self.

A fixation is the length of time the eye fixates on the area before the saccade occurs and is related to the information processing in a particular region. In addition, it is related to the time needed to plan subsequent saccades and the expected value of the information that will become available after the next saccade (*Brunyé et al., 2019*). The fixations can be used to infer fixation patterns through the examination of multiple metrics of fixations. The aim of the present study is to explore how self-criticism, specifically inadequate self and hated self, and other stimuli variables affect fixations on emotional faces.

## METHODS

### Participants

The research sample consisted of 120 adult participants from Slovakia, 60 women and 60 men; $M = 20.16$; $SD = 2.03$. We selected young respondents for the research sample as we wanted to explore the level of self-criticism, which peaks mostly in youth (see *e.g.*, *Fichman, Koestner & Zuroff, 1996*; *Neff & McGehee, 2010*). To conduct the multiple linear regression and therefore obtain a statistical power of 0.8 with 0.05 significance requires at least 27 participants (https://www.statskingdom.com/sample_size_regression.html). The highest scoring self-critical individuals (over 75 percentile in FSCRS subscale IS = inadequate self; $N = 44$; subscale HS = hated self; $N = 32$; IS + HS; $N = 48$) and the lowest scoring

self-critical individuals (under 25 percentile in IS; $N = 31$; in HS; $N = 42$; in IS + HS; $N = 26$) were selected from the sample for further analysis.

Participants were recruited from the general community by convenience sampling through social media using the snowball technique. Respondents were motivated to sign up for a session in the eye-tracking lab and a financial prize for participation was awarded to one individual at the end of the data collection by means of a prize draw. All procedures performed in this study were in accordance with the ethical standards of the institutional research committee and with the 1964 Helsinki declaration and its later amendments or comparable ethical standards. The study protocol was approved by the Ethical Committee of the Faculty of Social and Economic Sciences at Comenius University in Bratislava (approval number 2/2018). Informed consent was included in the online eye-tracking study and obtained from all individual participants at the beginning. The eye-tracking trial continued on-screen.

## Measures
### The Forms of Self-Criticism/reassuring Scale
This self-reporting questionnaire determines level of self-criticism. It consists of three subscales, two of them represent self-criticism: inadequate self and hated self, and a third subscale measures level of self-reassuring self. The questionnaire contains 22 statements, which participants evaluate on a scale of 0 to 4, with a value of 0 representing the statement "Not at all" and a value of 4 representing "Very much like me" (*Gilbert et al., 2004*). Previous results (*e.g.*, *Castilho, Pinto-Gouveia & Duarte, 2015*; *Kupeli et al., 2013*), including a Slovak sample (*Halamová, Kanovský & Pacúchová, 2017*), have shown that this scale has good reliability (Cronbach alfa = 0.75−0.85) and validity. The measure has also been validated in 13 different non-clinical samples, including a Slovak one (*Halamová et al., 2018*), suggesting a three-factored solution that distinguishes between inadequate, hated self, and reassured self.

### Eye-tracking metrics
Specific eye-tracking fixation metrics were chosen to measure the participants' eye movements during the free viewing task. These metrics were selected based on their relevance to the research aim and their capacity to provide information on the participants' visual attention and processing of facial expressions. The fixation metrics captured various aspects of fixation behaviour. The eye-tracking data was recorded using Tobii eye-trackers at a sampling rate of 60 Hz. The following dependent variables were the fixation metrics and were measured during the free viewing task:

1. Initial: This metric showed which facial expression (left or right) the participants had fixated on first. It provided information on the participants' initial attentional bias towards specific facial expressions (*Smith & Mital, 2015*).
2. Time to First Fixation (TTFF): TTFF, also known as First Fixation Latency, measured the time elapsed until first fixation on each type of facial expression in each trial. It represents the speed of initial attentional capture towards different facial expressions (*Hessels et al., 2016*).

3.  First Fixation Duration (FFD): FFD measured the duration of the first fixation on each type of facial expression in each trial. It indicated the participants' initial processing and engagement with different facial expressions (*Birmingham, Bischof & Kingstone, 2008*).

4.  Total Fixation Time (TFD): TFD measured the total time that each subject fixated on each type of facial expression in each trial. It provided an overall measure of visual attention towards different facial expressions (*Birmingham, Bischof & Kingstone, 2008*).

5.  Total Fixation Time in the Area of Interest (AOI) as % of Overall Exposure Time (TFD_ratio): TFD_ratio represented the total fixation time in the AOI, which was the region of interest containing the facial expressions, as a percentage of the overall exposure time. It indicated the relative attentional allocation towards the facial expressions compared to other visual stimuli (*Hessels et al., 2016*)

6.  Average Fixation Duration (AFD): AFD calculated the average duration of fixations on each type of facial expression. It provided insights into the participants' sustained attention and processing of different facial expressions (*Hessels et al., 2016*).

7.  Fixation Count (FC): FC counted the number of fixations made on each type of facial expression. It reflected the frequency of attentional shifts towards different facial expressions (*Smith & Mital, 2015*).

We examined potential differences in each metric between high and low self-critical respondents—on the inadequate self and hated self subscale—while participants were looking at different pairs of primary facial expressions and neutral expressions.

## Procedure

Once the written online consent form had been completed, the participants were seated in front of a computer screen. A pair of photos representing two of the six primary emotions (anger, disgust, fear, happiness, surprise, sadness) and a neutral emotion were then displayed on the screen. The stimulus set consists of static pictures of facial emotions taken from a standardised stimulus set—The Umeå University Database of Facial Expressions (*Samuelsson et al., 2012*). The procedure was conducted in real time, not online. Each pair (in total 126 colour images of pairs of facial expressions exhibited by the same model) appeared in the middle of a black screen for 5 s in random order. Before the pairs were shown, a black screen with a fixation cross in the middle appeared for 500 ms. Respondents were then instructed to look at the facial expressions with no further instructions—this was the free viewing task. Given the size of the screen (52.5 × 32.5 cm) and the respondent's distance from the screen (60 cm), the visual angle of the screen was 46.86°. We resized the photographs to simulate the real-life face-viewing process (see *Henderson, Williams & Falk, 2005*). All the photos in our research study were 5.8 cm × 8.7 cm (width × height), resolution 211 × 317 pixels. Areas of interest (AOIs) were defined manually as the entire facial expression in the picture, either on the left or right. After this viewing procedure, participants were asked to complete a self-reporting measure—The Forms of Self-Criticising/Attacking and Self-Reassuring Scale (FSCRS; *Gilbert et al., 2004*, translated into Slovak by *Halamová, Kanovský & Pacúchová, 2017*). The questionnaire was completed on a computer.

## Data analyses

We used IBM SPSS program version 26 (IBM Corp., Armonk, NY, USA) for the data analysis. Chi-square analyses were conducted to explore the differences in initial gaze between low and high self-critics depending on the emotion of the stimulus, sex of the stimulus, age of the stimulus, and a combination of these factors. Multiple linear regression was conducted to explore significant predictors in each dependent variable (Initial, TTFF, TFD total, TFD_ratio, FC, FFD, AFD).

## RESULTS

The results are presented in two separate sections based on the FSCRS subscale. In each section, the metrics are analysed by means of a dependent variables' comparison between high and low self-criticism and left and right visual field. For clarity, the results are presented separately for each dependent variable (inadequate and hated self). To provide a clear overview of the key findings, we have also included a summary table (Table 1) at the end of this results section. This table offers a concise comparison of significant fixation metrics, highlighting the influencing factors, explained variance ($R^2$), and the associated advantages and challenges. Readers are encouraged to refer to Table 1 for a quick and comprehensive understanding of the main outcomes discussed in detail below.

### Results for the inadequate self subscale
*1. Direction of initial gaze /Initial/*

In our data, there were no significant differences in initial gaze between low and high inadequate self (IS) self-critics.

(a) **Direction of initial gaze depending on type of stimulus emotion /Initial_E/**
No differences in preferred stimuli emotion were found in initial gaze between low and high IS self-critics ($\chi^2(6) = 2.993$; $p = .810$; V = .020).

(b) **Direction of initial gaze depending on sex of the stimulus /Initial_sex/**
No differences in preferred stimuli sex were found in initial gaze between low and high IS self-critics ($\chi^2(1) = .000$; $p = .509$; $\varphi = .000$).

(c) **Direction of initial gaze depending on age of the stimulus /Initial_age/**
No differences in preferred stimuli age were found in initial gaze between low and high IS self-critics ($\chi^2(2) = .000$; $p = 1.000$; V = .000).

(d) **Direction of initial gaze depending on emotion, sex and age of the stimulus /Initial_all/**
No differences in preferred combination of stimuli emotion, sex, and age were found in initial gaze between low and high IS self-critics ($\chi^2(41) = 19.413$; $p = .998$; V = .050).

*2. Time to first fixation /TTFF/*

Our results showed a significant effect of IS in TTFF in both the left and right visual field. Individuals scoring higher in IS took longer to reach first fixation.

***TTFF_L—Time to first fixation on stimuli presented in the left visual field.*** Results of the multiple linear regression indicated a significant effect of IS on TTFF_L, ($F(1, 7810) =$

**Table 1  Comparison of fixation metrics based on self-criticism subscales.**

| Metric | *p*-value | Influencing Factors | $R^2$ | Advantages | Limitations | Challenges |
|---|---|---|---|---|---|---|
| **Time to First Fixation (TTFF)** | $p < .001$ | IS | .007 | Indicates speed of attentional capture | Limited variance explained | Requires precise timing and data accuracy |
| TTFF_L | $p < .001$ | IS | .007 | Identifies lateral attentional bias | Small effect size; No effect of emotions, age, or sex | Lateral presentation may introduce bias |
| TTFF_R | $p < .001$ | IS | .007 | Identifies lateral attentional bias | Small effect size; No effect of emotions, age, or sex | Lateral presentation may introduce bias |
| **First Fixation Duration (FFD)** | $p < .001$ | IS | .002 | Measures initial engagement duration | Small effect size | Requires precise measurement; Potentially high variability |
| FFD_L | $p < .001$ | IS | .002 | Lateral differences can inform attentional processes | Small effect size; No effect of emotions, age, or sex | Ensuring consistent fixation durations |
| FFD_R | $p < .001$ | IS | .002 | Lateral differences can inform attentional processes | Small effect size; No effect of emotions, age, or sex | Ensuring consistent fixation durations |
| **Total Fixation Duration (TFD)** | $p < .001$ (TFD_L) $p = .002$ (TFD_R) | IS | .003 (TFD_L) .001 (TFD_R) | Comprehensive measure of visual attention | Small effect size; Mixed results for HS | Aggregating fixations accurately; Potential data noise |
| TFD_R | $p < .001$ | HS | .002 | Insight into sustained attention on right stimuli | Small effect size | Accurate aggregation of fixation data |
| **Total Fixation Duration Ratio (TFD_ratio)** | $p < .001$ | IS | .012 | Relative measure of attentional allocation | Small effect size | Ratio calculations can be complex |
| TFD_ratio_R | $p < .001$ | HS | .002 | Evaluates proportional attention | Small effect size | Accurate ratio computation |
| **Average Fixation Duration (AFD)** | $p < .001$ | IS | .005 | Indicates sustained attention on average | Small effect size | Variability in fixation duration |
| AFD_R | $p < .001$ | IS | .003 | Lateral differences in sustained attention | Small effect size; No effect of emotions, age, or sex | Ensuring consistent measurement |
| **Fixation Count (FC)** | $p < .001$(IS) $p = .015$ (sex) | IS Sex | .014 | Quantitative measure of attention shifts | Small effect size | Counting fixations accurately |
| FC_R | $p < .001$(HS) $p < .001$ (age) | HS Age | .002 | Demographic insights on attentional patterns | Small effect size | Accurate fixation count |

57.178, p <.001, $R^2$ = .007). Stimulus emotions, age, and sex had no significant effect on TTFF_L.

***TTFF_R—Time to first fixation on stimuli presented in the right visual field.*** Results of the multiple linear regression indicated a significant effect of IS on TTFF_R, $(F(1, 7810) = 55.945, p < .001, R^2 = .007)$. Stimulus emotions, age, and sex had no significant effect on TTFF_R.

Our results showed IS had a significant effect on TTFF in both the left and right visual field. Individuals scoring higher in IS took longer to reach first fixation.

For regression coefficients of TTFF_L and TTFF_R see Appendix A.

### 3. First fixation duration /FFD/

***FFD_L—First fixation duration on stimuli presented in the left visual field.*** Results of the multiple linear regression indicated a significant effect of IS on FFD_L, $(F(1, 7810) = 17.502, p < .001, R^2 = .002)$. Stimulus emotions, age, and sex had no significant effect on FFD_L.

***FFD_R—First fixation duration in stimuli presented in the right visual field.*** Results of the multiple linear regression indicated a significant effect of IS on FFD_R, $(F(1, 7810) = 11.933, p < .001, R^2 = .002)$. Stimulus emotions, age, and sex had no significant effect on FFD_R.

IS had a significant effect on FFD in both the left and right visual field. Individuals scoring higher in IS had a longer first fixation.

For regression coefficients of FFD_L and FFD_R see Appendix B.

### 4. Total fixation duration /TFD/ - L/R

***TFD_L—Total fixation duration on stimuli presented in the left visual field.*** Results of the multiple linear regression indicated a significant effect of IS on TFD_L, $(F(1, 7810) = 23.89, p < .001, R^2 = .003)$. Stimulus emotions, age, and sex had no significant effect on TFD_L.

***TFD_R—Total fixation duration on stimuli presented in the right visual field.*** Results of the multiple linear regression indicated a significant effect of IS on TFD_R, $(F(1, 7810) = 9.95, p = .002, R^2 = .001)$. Stimulus emotions, age, and sex had no significant effect on TFD_R.

Our results showed a significant effect of IS in TFD in both the left and right visual field. All the fixations take longer time in individuals scoring higher in IS.

For regression coefficients of TFD_L and TFD_R see Appendix C.

### 5. Total Fixation Duration Ratio

***TFD_ratio_L—total fixation duration ratio on stimuli presented in the left visual field.*** Results of the multiple linear regression indicated a significant effect of IS on TFD_ratio_L, $(F(1, 7810) = 93.638, p < .001, R^2 = .012)$. Stimulus emotions, age, and sex had a significant effect on TFD_ratio_L.

***TFD_ratio_R—Total Fixation Duration Ratio on stimuli presented in the right visual field.***

Results of the multiple linear regression indicated a significant effect of IS on TFD_ratio_R, $(F(1, 7810) = 10.0$, $p = .002$, $R^2 = .001)$. Stimulus emotions, age and sex had no significant effect on TFD_ratio_R.

Our results showed IS had a significant effect on TFD_ratio in both the left and right visual field.

For regression coefficients of TFD_ratio_L and TFD_ratio_R see Appendix D.

### 6. Average Fixation Duration /AFD/

***AFD_L—Average fixation duration on stimuli presented in the left visual field.*** Results of the multiple linear regression indicated a significant effect of IS on AFD_L, $(F(1, 7810) = 37.01$, $p < .001$, $R^2 = .005)$. Stimulus emotions, age, and sex had no significant effect on AFD_L.

***AFD_R—Average fixation duration on stimuli presented in the right visual field.***
Results of the multiple linear regression indicated a significant effect of IS on AFD_R, $(F(1, 7810) = 19.837$, $p < .001$, $R^2 = .003)$. Stimulus emotions, age, and sex had no significant effect on AFD_R.

Our results showed IS had a significant effect on AFD in both the left and right visual field. Fixations of individuals scoring higher in IS lasted longer on average.

For regression coefficients of AFD_L and AFD_R see Appendix E.

### 7. Fixation Count /FC/

***FC_L—Fixation count for stimuli presented in the left visual field.*** Results of the multiple linear regression indicated significant effects of IS and sex of the stimulus on FC_L, $(F(1, 7810) = 114.256$, $p < .001$, $R^2 = .014)$. Both IS (B = .627; $p < .001$) and sex of the stimulus (B = −.142; $p = .015$) contributed significantly to the model. Stimulus emotions and age did not have a significant effect on FC_L.

***FC_R—Fixation count for stimuli presented in the right visual field.*** Results of the multiple linear regression indicated a significant effect of IS on FC_R, $(F(1, 7810) = 27.848$, $p < .001$, $R^2 = .004)$. Stimulus emotions, age, and sex had no significant effect on FC_R.

Our results showed a significant effect of IS on FC in both the left and right visual field. Individuals scoring higher in IS had more fixations. In the left visual field, the sex of the stimulus also had a significant effect. Individuals scoring higher in IS fixated more on male faces.

For regression coefficients of FC_L and FC_R see Appendix F.

## Results for the hated self subscale
### 1. Direction of initial gaze /Initial/

In our data, there were no significant differences in initial gaze between low and high hated self (HS) self-critics.

(a) **Direction of initial gaze depending on type of stimulus emotion /Initial_E/**

No differences in preferred stimuli emotion were found in initial gaze between low and high HS self-critics ($\chi^2(6) = 1.360$; $p = .968$; V $= .012$).

(b) **Direction of initial gaze depending on the sex of the stimulus /Initial_sex/**
No differences in preferred stimuli sex were found in initial gaze between low and high HS self-critics ($\chi^2(1) = .000$; $p = .986$; $\varphi = .000$).

(c) **Direction of initial gaze depending on age of the stimulus /Initial_age/**
No differences in preferred stimuli age were found in initial gaze between low and high HS self-critics ($\chi^2(2) = .000$; $p = 1.000$; V $= .000$).

(d) **Direction of initial gaze depending on emotion, sex, and age of the stimulus /Initial_all/**
No differences in preferred combination of stimuli emotion, sex, and age were found in initial gaze between low and high HS self-critics ($\chi^2(41) = 7.578$; $p = 1.000$; V $= .029$).

### *2. Time to first fixation /TTFF/*

***TTFF_L—Time to first fixation on stimuli presented in the left visual field.*** Results of the multiple linear regression indicated no significant effect of HS, stimulus emotions, age, or sex on TTFF_L, ($F_{(4, 9317)} = .590$, $p = .670$, $R^2 = .000$).

***TTFF_R—Time to first fixation on stimuli presented in the right visual field.*** Results of the multiple linear regression indicated no significant effect of HS, stimulus emotions, age, or sex on TTFF_R, ($F_{(4, 9317)} = .105$, $p = .981$, $R^2 = .000$).
For regression coefficients of TTFF_L and TTFF_R see Appendix G.

### *3. First fixation duration /FFD/*

***FFD_L—First fixation duration on stimuli presented in the left visual field.*** Results of the multiple linear regression indicated no significant effect of HS, stimulus emotions, age, or sex on FFD_L, ($F_{(4, 9317)} = .307$, $p = .847$, $R^2 = .000$).

***FFD_R—First fixation duration on stimuli presented in the right visual field.*** Results of the multiple linear regression indicated no significant effect of HS, stimulus emotions, age, or sex on FFD_R, ($F_{(4, 9317)} = .768$, $p = .546$, $R^2 = .000$). For regression coefficients of FFD_L and FFD_R see Appendix H.

### *4. Total fixation duration /TFD/ - L/R*

***TFD_L—Total fixation duration on stimuli presented in the left visual field.*** Results of the multiple linear-regression indicated no significant effect of HS, stimulus emotions, age, or sex on TFD_L, ($F_{(4, 9317)} = .851$, $p = .493$, $R^2 = .000$).

***TFD_R—Total fixation duration on stimuli presented in the right visual field.*** Results of the multiple linear regression indicated a significant effect of HS on TFD_R, ($F_{(4, 9317)} = 5.132$, $p = .000$, $R^2 = .002$). Stimulus emotions, age, and sex had no significant effect on TFD_R.

Our results showed HS had a significant effect on TFD in the right visual field only. Individuals scoring higher in HS took longer to fixate on objects in the right visual field. For regression coefficients of TFD_L and TFD_R see Appendix I.

### 5. Total Fixation Duration Ratio

***TFD_ratio_L—Total Fixation Duration Ratio on stimuli presented in the left visual field.*** Results of the multiple linear regression indicated no significant effect of HS, stimulus emotions, age or sex on TFD_ratio_L, ($F(4, 9317) = 1.921$, $p = .104$, $R^2 = .001$).

***TFD_ratio_R—Total Fixation Duration Ratio on stimuli presented in the right visual field.*** Results of the multiple linear regression indicated a significant effect of HS on TFD_ratio_R, ($F(4, 9317) = 5.103$, $p = .000$, $R^2 = .002$). Stimulus emotions, age, and sex had no significant effect on TFD_ratio_R.

Our results showed HS had a significant effect on TFD_ratio in the right visual field only. Individuals scoring higher in HS took longer to fixate on objects in the right visual field.

For regression coefficients of TFD_ratio_L and TFD_ratio_R see Appendix J.

### 6. Average Fixation Duration /AFD/

***AFD_L—Average fixation duration on stimuli presented in the left visual field.*** Results of the multiple linear regression indicated no significant effect of HS, stimulus emotions, age, or sex on AFD_L, ($F(4, 9317) = 1.535$, $p = .189$, $R^2 = .002$).

***AFD_R—Average fixation duration on stimuli presented in the right visual field.*** Results of the multiple linear regression indicated no significant effect of HS, stimulus emotions, age, or sex on AFD_R, ($F(4, 9317) = 1.015$, $p = .398$, $R^2 = .000$). For regression coefficients of AFD_L and AFD_R see Appendix K.

### 7. Fixation Count /FC/

***FC_L—Fixation count on stimuli presented in the left visual field.*** Results of the multiple linear regression indicated sex of the stimulus had a significant effect on FC_L, ($F(4, 9317) = 3.966$, $p < .003$, $R^2 = .002$). HS, stimulus emotions, and age did not have a significant effect on FC_L.

***FC_R—Fixation count on stimuli presented in the right visual field.*** Results of the multiple linear regression indicated that HS and age of the stimulus had significant effects on FC_R, ($F(4, 9317) = 4.535$, $p < .001$, $R^2 = .002$). Stimulus emotions and sex had no significant effect on FC_R.

Our results showed significant effect of HS in FC in both visual fields. For individuals scoring higher in HS, the fixation count was higher.

For regression coefficients of FC_L and FC_R see Appendix L.

To sum up the results, Table 1 presents a summarised comparison of the results based on various significant fixation metrics, highlighting the influencing factors, explained variance ($R^2$), advantages, drawbacks, and challenges associated with each significant finding.

## DISCUSSION

In this research study, we analysed fixation patterns of self-critical individuals in relation to level of self-criticism. In particular, we tested to see whether there were any differences in onset and duration of fixation on emotional faces in high *versus* low self-critical participants, identified using both subscales of self-criticising self by *Gilbert et al. (2004)*—inadequate and hated self. The findings revealed no differences in initial gaze between low and high self-critics. However, self-criticism had a significant effect on time to first fixation, duration of first fixation, and total fixation duration, as well as on average fixation duration (regardless of whether the stimulus was the left or right of the pair).

In those with a higher inadequate self score, the metrics of first fixation—specifically, time to first fixation and duration of the first fixation—decreased, which means that onset of gaze occurred earlier in more self-critical participants exhibiting inadequate self or with a shorter delay than for respondents with lower inadequate self.

Furthermore, in the pattern of total scanning of primary emotional expressions, respondents with a higher level of inadequate self exhibited a sustained increase in fixation metrics, including total fixation duration, total fixation duration ratio, average fixation duration, and fixation count. This pattern suggests that respondents with beliefs typical of the inadequate self may be more prone to prolonged scrutiny of emotional expressions. These individuals might believe they deserve self-criticism because they perceive others as superior, highlighting their own imperfections (*Gilbert et al., 2004*). Consequently, they tend to fixate more quickly on emotional expressions and also maintain their focus for longer periods as they scrutinize the expression in detail.

The results are in line with previous findings that individuals with higher pathological self-criticism—hated self—had higher avoidance of the eye region when looking at happy facial expressions, to which self-critical people are very sensitive (*McEwan et al., 2014*). However, for inadequate self, previous results did not confirm an avoidance tendency when scanning or recognising emotional faces, or vice versa (*Strnádelová, Halamová & Mentel, 2019*). Inadequate self is a self-critical form and tended to correlate positively with total fixation duration time on all areas examined on emotional faces –eyes, around the eyes, and the mouth (*Strnádelová, Halamová & Mentel, 2019*). The present results show that the tendency to avoid first contact (fixation) was not shown with the whole expression among highly self-critical people.

The results are consistent with previous findings that individuals with higher levels of pathological self-criticism—specifically, the hated self—tend to avoid the eye region when viewing happy facial expressions, a sensitivity common among self-critical individuals (*McEwan et al., 2014*). This avoidance behavior aligns with our observation that those with high hated self experience difficulties in scanning and recognizing emotional faces, particularly when the task requires rapid processing within a limited time frame (*Strnádelová, Halamová & Kanovský, 2019*). These individuals exhibit longer total fixation durations and a higher fixation count, which may indicate either a struggle with adequate expression scanning or difficulty in detecting the emotional expression, leading to prolonged fixation on specific facial areas.

In contrast, the results related to inadequate self differ from those of previous studies. While *Strnádelová, Halamová & Mentel (2019)* found no significant avoidance tendency in individuals with high inadequate self when scanning or recognizing emotional faces, our findings suggest that these individuals may actually exhibit increased attention towards emotional expressions. Specifically, we observed that individuals with higher inadequate self showed a positive correlation with total fixation duration on all emotional faces, indicating a heightened level of scrutiny. This is in line with the idea that inadequate self, as a form of self-criticism, involves a more general vigilance and concern about personal shortcomings, leading to prolonged examination rather than avoidance.

Moreover, the present study highlights that while individuals with high hated self demonstrate an attention avoidance tendency, especially in tasks with time constraints (*McEwan et al., 2014*), those with higher inadequate self tend to have an earlier onset of gaze and longer fixation durations when viewing pairs of emotional facial expressions. This suggests that inadequate self may drive a different pattern of emotional processing, where the focus is on detailed scrutiny rather than avoidance, contrasting with the patterns observed in those with high hated self or a combined self-criticism score (*Halamová et al., 2023*). Therefore, the current findings provide a nuanced understanding of how different forms of self-criticism—hated self *versus* inadequate self—affect emotional face processing.

With regard to fixation count, in addition to the significant effect of self-criticism (inadequate and hated self), there was a significant effect of sex of the stimulus in left position (FC_L). Participants exhibited significantly more fixations when a male stimulus was presented in the left visual field. They looked back and had more fixations on male stimuli when they were positioned on the left of the screen, probably because of the stimulus characteristics. *Radlow (2019)* in her thesis found that horizontal eye movements were more frequently connected to stimuli-characteristics, liking and complexity than to participant personality variables. We did not evaluate the stimuli characteristics. Such an evaluation could be a future research direction—analysing personal characteristics and stimuli characteristics such as likeability or attractivity.

One notable limitation of this psychological eye-tracking study is sample size and composition. Our study contained 120 participants, a number that, while adequate for initial exploratory analysis, may not provide a fully representative cross-section of the general population. This limitation affects the generalisability of our findings in several key ways. For example, the sample may lack sufficient demographic diversity in terms of age, ethnicity, socioeconomic status, and educational background. The absence of a balanced representation of these demographic variables could mean that our results are skewed or fail to capture the full spectrum of human variability in eye-tracking responses and psychological phenomena. A sample of 120 participants may not be sufficient to capture the breadth of behavioural variability seen in larger, more diverse populations. Psychological responses and eye-tracking patterns can differ significantly across different groups and contexts (*Holmqvist et al., 2011*; *Rayner, 2009*), and a limited sample size restricts our ability to generalise these findings to broader populations.

Despite the limitations of the study, and considering the fixation count metrics, several variables appeared to influence the results, including the sex of the stimulus in the left

position, age in the right position (FC_R), and self-criticism (both inadequate and hated self) in either position. However, for the other eye-tracking metrics—such as total fixation duration and time to first fixation, self-criticism was consistently found to have a significant effect, whereas variables like age or sex of the stimulus did not demonstrate a similar impact. This suggests that fixation count may not be as reliable or sensitive a metric for predicting self-criticism when compared to these other metrics. Fixation count alone might be influenced by various factors unrelated to self-criticism, thereby making it less specific in capturing the nuances of self-critical processing. In contrast, metrics like total fixation duration and time to first fixation provide a more robust reflection of the underlying cognitive and emotional processes associated with self-criticism.

The biggest contribution of this research is that it is the first study to consider all the primary emotions and neutral expressions of a stimulus and observe the differences in eye-tracking of facial expressions. Previous research procedures have focused on analysing a particular emotion—for example mainly sad/happy faces for depressed people (*e.g., Allard & Yaroslavsky, 2019*; *Sanchez et al., 2013*), or sad/angry/happy faces for anxious individuals (*Armstrong & Olatunji, 2012*; *Horley et al., 2003*), or selected eye-tracking metrics of fixations, mainly total fixation duration (*Halamová et al., 2023*; *Strnádelová, Halamová & Mentel, 2019*; *Strnádelová, Halamová & Kanovský, 2019*).

The differentiation between inadequate self and hated self is critical for several reasons, both theoretically and practically. These two forms of self-criticism represent distinct psychological constructs that manifest differently in emotional processing and behavioral patterns (*Gilbert et al., 2004*; *Whelton & Greenberg, 2005*). Understanding these distinctions is essential for tailoring psychological interventions more effectively. For example, individuals with high levels of inadequate self may benefit from interventions that focus on enhancing self-compassion and reducing excessive self-monitoring (*Gilbert, 2010*; *Neff & Germer, 2013*), whereas those with high levels of hated self may require strategies aimed at reducing self-hatred and improving emotional regulation to decrease avoidance behaviors (*Leary et al., 2007*). By identifying the specific form of self-criticism a person struggles with, therapists can implement more targeted and cost-effective interventions, potentially leading to better outcomes (*Kirschner et al., 2019*). This nuanced understanding could help refine therapeutic approaches and improve their efficacy, justifying the initial investment in eye-tracking technology. Thus, the differentiation between inadequate self and hated self is not only important for advancing our theoretical understanding of self-criticism but also has practical implications for the development of more personalized and effective treatments.

## Future work

Future research could benefit from including a more diverse sample, encompassing a wider age range as well as different cultural and social contexts, and a variety of clinical diagnoses. Prior studies have shown that age, cultural background, and clinical condition significantly affect attentional and emotional processing (*Isaacowitz et al., 2006*). A more heterogeneous sample would enhance the generalisability of the findings and provide a

more comprehensive understanding of these processes across different populations (*Smith & Mital, 2015*).

It would also be worth assessing additional variables that may influence self-criticism and emotional processing, such as personality traits, adverse life experiences, or levels of anxiety and depression. Research has demonstrated that personality traits like neuroticism and extraversion, as well as adverse life experiences, can significantly affect emotional regulation and attentional biases (*Matthews & Wells, 1996*; *Williams, Mathews & MacLeod, 1996*). Including these variables would provide a more nuanced understanding of the factors that contribute to self-criticism and its impact on emotional processing.

Last, but not least, conducting longitudinal studies that follow participants over time following interventions aimed at reducing self-criticism could provide important evidence on the effectiveness of these interventions in changing not only self-perception but also emotional processing patterns. Longitudinal research is critical for understanding the long-term effects of psychological interventions and for identifying the mechanisms through which these interventions have an effect (*Kazdin, 2016*). Studies have shown that interventions can lead to lasting changes in both self-criticism and emotional processing (*Halamová et al., 2023*; *Kroener, Mahler & Sosic-Vasic, 2023*). Therefore, longitudinal designs would be valuable for assessing the durability and impact of these interventions over time.

## CONCLUSION

Compared with previous research findings showing an attention bias for negative stimuli or positive stimuli (*e.g.*, *Armstrong & Olatunji, 2012*; *Allard & Yaroslavsky, 2019*) in clinical samples with a diagnosis closely related to self-criticism, the current results showed that highly self-critical people with a higher level of inadequate self and hated self exhibited significant sustained attention in total fixation scanning of all primary facial emotions and also had an earlier onset of first fixation among participants with a higher level of inadequate self. The results contribute to a better understanding of how eye-tracking processes can help us distinguish between different forms of self-criticism (inadequate and hated) and various fixation metrics for detecting self-criticism as another potential marker of psychopathology.

## ACKNOWLEDGEMENTS

We would like to acknowledge Lenka Lysá, Silvia Pukanová, Karolína Šandalová, Dominika Šoltésová, Silvia Štellerová and Alexandra Vrábelová for the help with data gathering.

### Funding

This work was supported by the Slovak Research and Development Agency under the Contract no. PP-COVID-20-0074. Writing this work was supported by the Vedecká grantová agentúra VEGA under Grant VEGA 1/0075/19 and VEGA 1/0725/19. The funders

had no role in study design, data collection and analysis, decision to publish, or preparation of the manuscript.

### Grant Disclosures

The following grant information was disclosed by the authors:

The Slovak Research and Development Agency: PP-COVID-20-0074.

The Vedecká grantová agentúra VEGA: VEGA 1/0075/19, VEGA 1/0725/19.

### Competing Interests

The authors declare there are no competing interests.

### Author Contributions

- Bronislava Šoková conceived and designed the experiments, performed the experiments, prepared figures and/or tables, authored or reviewed drafts of the article, and approved the final draft.
- Martina Baránková analyzed the data, prepared figures and/or tables, authored or reviewed drafts of the article, and approved the final draft.
- Júlia Halamová conceived and designed the experiments, authored or reviewed drafts of the article, and approved the final draft.

### Ethics

The following information was supplied relating to ethical approvals (*i.e.*, approving body and any reference numbers):

Ethical Committee of the Faculty of Social and Economic Sciences at Comenius University in Bratislava (approval number 2/2018).

### Data Availability

The raw measurements, including fixations, are available in the Supplementary Files.

### Supplemental Information

Supplemental information for this article can be found online at http://dx.doi.org/10.7717/peerj-cs.2413#supplemental-information.

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
