# Peer review of "Fixation patterns in pairs of facial expressions—preferences of self-critical individuals"

_PeerJ Computer Science, doi:10.7717/peerj-cs.2413_

## Round 0.1 · original submission · Major Revisions

The Reviewers agree on the interestingness of the topic. However, the Authors are required to take some actions to better deliver their work and allow a better readability of the paper.

The Authors must check the formatting of the paper and provide a professional revision of English language writing.

The investigated problem must be clarified, and the contributions clearly stated. The methodology description lacks a clear writing and organization and requires to be reported in more detail.

Please, answer the comments raised by the Reviewers concerning this matter as well as considering a better reporting of the results.

Please, revise attentively the discussed findings, the limitations of the study, and the conclusions by providing a more structured and detailed text.

Reviewer 1 ·

Basic reporting

It is suggested to briefly introduce the background and challenge in an abstract and expound more on the detailed work.

Experimental design

no comment

Validity of the findings

It would be better to summarize the results in a table (comparisons in terms of complexity, advantages, drawbacks, challenges, or limitations, etc.) for easier understanding.

·

Basic reporting

No comments

Experimental design

No comments

Validity of the findings

No comments

Additional comments

This paper presents Fixation patterns in the pairs of facial expression preferences of self-critical individuals. The results contribute to a better understanding of eye-tracking processes distinguishing between different forms of self-criticism (inadequate and hated) and various fixation metrics for the detection of self-criticism as another potential marker of psychopathology.

The article is well-written and detailed, focusing on a specific and underexplored area. The main positive points are:

1. The work is a significant contribution to the Understanding of Self-Criticism. The study provides valuable insights into how different forms of self-criticism influence the visual processing of emotional facial expressions.

2. The study's results have profound practical and clinical implications. By identifying specific patterns of visual attention associated with self-criticism, potential pathways for therapeutic interventions are suggested, which may help mitigate the harmful effects of self-criticism.

3. Eye Tracking provides a window to understand underlying cognitive processes non-invasively and in real time.

However, some points need to be adjusted:

1. In the methodology, as well as in the entire article, I missed a model architecture, or a big picture providing an overview of the methodology. Also, in the methodology section, there was a lack of a clearer division of topics to clarify where it begins and ends. Without this, the reading was a bit confusing.

2. In the section "Self-criticism and eye-tracking," I felt a lack of deeper analysis between "Inadequate Self" and "Hated Self." Further investigations on how each uniquely influences the perception and processing of facial expressions could provide deeper insights.

3. A notable limitation of the study lies in its sample of 120 people, which may not be widely representative of the general population, thus limiting the generalizability of the results.

4. The article could benefit from a more detailed discussion on how specific eye-tracking metrics were chosen and interpreted in the context of the study's hypotheses, which would fit into the methodology section. Furthermore, deepening the statistical analysis to explore possible confounding variables or interactions between variables could strengthen the conclusions drawn.

5. There is nothing in the article about future work. Based on the observations and limitations previously discussed, I would recommend the following directions:

- Sample Expansion: Future research could benefit from including a more diverse sample, encompassing a wider age range, different cultural and social contexts, and individuals with various clinical diagnoses.

- Investigation of Additional Psychological Variables: It would be valuable to incorporate the assessment of additional variables that may influence self-criticism and emotional processing, such as personality traits, adverse life experiences, and levels of anxiety and depression.

- Longitudinal and Interventions: Implementing longitudinal studies that follow participants over time after interventions aimed at reducing self-criticism could provide important evidence on the effectiveness of these interventions in changing not only self-perception but also emotional processing patterns.

Reviewer 3 ·

Basic reporting

This manuscript presents findings from an eye tracking experiment designed to examine the relationship between fixation patterns and the personality/emotional trait of self-criticism. Before submission for peer-review, extensive editing would be necessary to enhance clarity, grammar, and reporting style to meet professional standards. Additionally, numerous formatting errors are evident, such as inconsistent line spacing (observed between lines 140-150).

The inclusion of literature references adequately establishes field background/context, though substantial improvements are needed in style: for instance, replacing colloquial phrases like "talked about" with more formal language such as "discussed" or "analyzed" (line 132) would enhance professionalism. Additionally, phrases like "we mentioned that" (line 121) can be refined to "it was demonstrated" or "the authors indicated."

Regarding conceptual formulation, closer attention is needed in articulating the motivation behind the study (lines 156-163, 179-183). In contrast, the section outlining the study's aims (lines 184-195) demonstrates clear writing, and can serve as a model for improving the manuscript overall.

Overall, the article structure, figures, tables, and sharing of raw data require significant refinement. The presentation of results is disjointed, resembling a preliminary lab report rather than a coherent presentation of findings. Statistical results are scattered and lack integration into a cohesive narrative.

Experimental design

The study likely falls within the scope of the journal, and the posed question holds significance. However, due to deficiencies in writing, the identified knowledge gap remains inadequately articulated and discernible to the reader.

The investigation upholds reasonably good technical and ethical standards. Nevertheless, the methods lack sufficient details.
To mention a few examples: 1) the rationale behind the power calculation, specifying the necessity of 24 participants for conducting multiple regression, is unclear.
2)There is ambiguity regarding the execution of parts of the experiment, with a mix of online and lab-based procedures, which was not explicitly delineated.
3) In line 245, the exact definition of areas of interest remains ambiguous.
4) Additionally, in lines 253-262, a more thorough explanation of metrics is warranted, particularly regarding the definition of fixations and eye movements. This section appears more akin to a lab report than a scientific manuscript.

Validity of the findings

The Results section suffers from poor writing, resembling an initial draft of a lab report, with a mere listing of statistical test outcomes. Crucially, the section lacks cohesion, presenting as a disjointed compilation of statistical tests.

A suggested improvement involves writing the results section as a coherent text revolving around the main hypotheses rather than listing the statistical tests. Then results tables could be moved from the appendix into the main text, while highlighting significant findings in bold. In scientific journals, Results are written similar to a narrative text and not as a list of statistical tests. Currently, the section lacks readability and coherence.

Following thorough revision of preceding sections, it is recommended that the Discussion be structured coherently and logically, with clear subheadings corresponding to explicitly stated hypotheses.

Additional comments

This study presents intriguing findings, yet it requires meticulous editing before submission for peer review. The current level of inaccuracies in this preliminary draft would likely prevent its acceptance for publication in peer-reviewed journals.

---

## Round 0.2 · Minor Revisions

I thank the Authors for submitting a revised version of their manuscript and for answering the Reviewers’ comments, following thoroughly the given guidelines.

To further improve the manuscript, please follow the additional comments provided by the Reviewers, by being particularly attentive on the content delivery, the abstract preparation, the presentation of the results, the clarification of the discussion, and the justification of some strong statements that could impede a clear understanding of your findings.

I suggest including in the section devoted to the results a more descriptive paragraph to summarise and comment on the reported point-by-point results. Consider exploiting the already added Table 1 to provide a clear assessment of the various observations you have previously given.

·

Basic reporting

no comment

Experimental design

no comment

Validity of the findings

no comment

Additional comments

The article meets the PeerJ criteria and the requested requirements were met. Therefore the article will be accepted.

Reviewer 3 ·

Basic reporting

Authors have significantly improved the basic reporting after the first revision. There are some minor issues to be addressed before this work can be accepted.

Experimental design

Well-done, no comment.

Validity of the findings

Findings are valid but the presentation and discussion can still be improved (see under 4).

Additional comments

The revised manuscript has improved a lot, however, the writing can be still improved, particularly in order to make the text coherent, comprehensible and clear. Below are a few examples (please note that this list is not exhaustive; the authors are encouraged to thoroughly review the manuscript and address other areas that may require improvement):

1) Abstract: instead of using emotional words such as concerning or worrying clearly state the scientific problem and why that is important: "So far, studies have revealed some worrying aspects of eye-tracking behaviours among self-critical individuals".

2) Abstract: clarify this statement, separating the study material to survey and behavioural tasks (stimuli) and then mention the equipment: "We used The Forms of Self-Criticizing and Self-Reassuring Scale to measure self-criticism and the Tobii X2 eye tracker, and The Umeå University Database of Facial Expressions Database as the stimulus for eye-tracking emotions.".

3) Abstract: "Results showed that in highly self-critical participants with Inadequate Self time to first fixation and duration of first fixation was shorter" you need a comma after Inadequate Self, otherwise this sentence is hard to parse.

4) Abstract coherence: Please make sure that the abstract succinctly conveys the essence of the study. I would spell out that based on the surveys, two groups were defined and the pattern of fixations and eye movements were compared. Then succinctly present the findings (similar to how a summary is provided at the end of the Introduction page 7- and 8).

5) Introduction: much better than the previous version. Pay attention to typos, e.g. word "investigation" page 7.

6) Thank you for providing a clear list of the metrics used to characterise the fixations and eye movements in Methods. To make this even better, could you rephrase: "We examined potential differences in each metric between high and low self-critical respondents – on the Inadequate Self and Hated Self subscale – while participants were looking at different pairs of primary facial expressions and neutral expressions.", in page 10?

7) Results: multiple typos here, e.g. page 13 all fixations takes. For me, reading the results section of this manuscript is almost impossible. The summary table (Table 1), where measured metrics are listed and the significant effects are highlighted is an extremely helpful addition and I suspect this would be the only thing that the reader would be able to appreciate. Please refer to this table much earlier in the text and before a reader potentially loses interest to read further. Please also clarify some of the metrics such as TTFF_L and TTFF_R (referring to left and right).

8) Discussion: Please improve the clarity of first paragraph, page 20 "That suggests it may occur in respondents experiencing beliefs typical of inadequate self." Also in the same page, when discussing previous research, please link the current findings with those more coherently, e.g. it is not clear why the results of (Strnádelová et al., 2019b) are discussed here, are the current results in line with those, or are in contrast with those, or those findings offer an explanation for the current findings? Please consider revising the Discussion section to clearly articulate how your findings relate to those of previous studies.

9) I am not sure about this statement :"However, for the other metrics, self-criticism was found to have a significant effect – rather than other variables such as age or sex of the stimulus. That could mean that fixation count is not a reliable eye-tracking metric for predicting self-criticism compared to others.". why fixation count is not a reliable metric does not follow from the preceding text (based on the results presented in the two last rows of Table 1).

10) After reading this manuscript, I am unclear about the importance of distinguishing between the two forms of self-criticism. Would this differentiation have any impact on the type of intervention used? Is it cost-effective, considering that eye-tracking is not an inexpensive method for assessing participants? Please consider adding a concise statement addressing these points in the Discussion section.

---

## Round 0.3 · accepted · Accept

I thank the Authors for having thoroughly followed and answered the Reviewers’ comments. The paper is now ready for publication.

The Authors have improved content delivery, the abstract, and the sections dedicated to the results and their discussion.

Reviewer 3 ·

Basic reporting

OK

Experimental design

OK

Validity of the findings

OK

Additional comments

The authors satisfactorily addressed my concerns.